# Regulatory Peptide Pro-Gly-Pro Accelerates Neuroregeneration of Primary Neuroglial Culture after Mechanical Injury in Scratch Test

**DOI:** 10.3390/ijms252010886

**Published:** 2024-10-10

**Authors:** Zanda Bakaeva, Mikhail Goncharov, Fyodor Frolov, Irina Krasilnikova, Elena Sorokina, Arina Zgodova, Elena Smolyarchuk, Sergey Zavadskiy, Liudmila Andreeva, Nikolai Myasoedov, Andrey Fisenko, Kirill Savostyanov

**Affiliations:** 1National Medical Research Center of Children’s Health, 119296 Moscow, Russia; irinakrsl81@gmail.com (I.K.); sorokelena@mail.ru (E.S.);; 2I.M. Sechenov First Moscow State Medical University (Sechenov University), 119991 Moscow, Russia; fyodorfrolov297@gmail.com (F.F.); smolyarchuk_e_a@staff.sechenov.ru (E.S.);; 3Kalmyk State University Named after B.B. Gorodovikov, 358000 Elista, Russia; 4Institute of Immunology, Christian-Albrechts-University of Kiel and University Medical Center Schleswig-Holstein, 24105 Kiel, Germany; misha121097@gmail.com; 5National Research Centre «Kurchatov Institute» (NRC «Kurchatov Institute»), 123182 Moscow, Russia; andr-la.img@yandex.ru (L.A.); myasoedov-nf.img@yandex.ru (N.M.)

**Keywords:** traumatic brain injury, delayed calcium deregulation, primary neuroglial cultures, cell scratch assay, Pro-Gly-Pro, neuroprotection

## Abstract

The scratch test is used as an experimental in vitro model of mechanical damage to primary neuronal cultures to study the mechanisms of cell death in damaged areas. The involvement of NMDA receptors in processes leading to delayed neuronal death, due to calcium dysregulation and synchronous mitochondrial depolarization, has been previously demonstrated. In this study, we explored the neuroregenerative potential of Pro-Gly-Pro (PGP)—an endogenous regulatory peptide with neuroprotective and anti-inflammatory properties and a mild chemoattractant effect. Mechanical injury to the primary neuroglial culture in the form of a scratch caused acute disruption of calcium homeostasis and mitochondrial functions. This was accompanied by neuronal death alongside changes in the profile of neuronal markers (BDNF, NSE and GFAP). In another series of experiments, under subtoxic doses of glutamate (Glu, 33 μM), delayed changes in [Ca^2+^]_i_ and ΔΨm, i.e., several days after scratch application, were more pronounced in cells in damaged neuroglial cultures. The percentage of cells that restored the initial level of [Ca^2+^]_i_ (*p* < 0.05) and the rate of recovery of ΔΨm (*p* < 0.01) were decreased compared with undamaged cells. Prophylactic application of PGP (100 μM, once) prevented the increase in [Ca^2+^]_i_ and the sharp drop in mitochondrial potential [ΔΨm] at the time of scratching. Treatment with PGP (30 μM, three or six days) reduced the delayed Glu-induced disturbances in calcium homeostasis and cell death. In the post-glutamate period, the surviving neurons more effectively restored the initial levels of [Ca^2+^]_i_ (*p* < 0.001) and Ψm (*p* < 0.0001). PGP also increased intracellular levels of BDNF and reduced extracellular NSE. In the context of the peptide’s therapeutic effect, the recovery of the damaged neuronal network occurred faster due to reduced astrogliosis and increased migration of neurons to the scratch area. Thus, the peptide PGP has a neuroprotective effect, increasing the survival of neuroglial cells after mechanical trauma in vitro by reducing cellular calcium overload and preventing mitochondrial dysfunction. Additionally, the tripeptide limits the post-traumatic consequences of mechanical damage: it reduces astrogliosis and promotes neuronal regeneration.

## 1. Introduction

Traumatic brain injury (TBI) causes a cascade of pathological changes within the axon such as abnormal ionic homeostasis that leads to axonal disconnection and necrotic or apoptotic cell death [1,2]. These processes cause secondary neuronal damage [3,4], which detrimentally impacts cellular viability in a long-term perspective [5,6]. The post-traumatic pathological processes, which underlie this condition, include cerebral blood flow dysregulation, glutamate excitotoxicity and neuroinflammation [7,8,9]. Together, these processes create a positive feedback loop, spreading throughout the brain and damaging nervous tissue adjacent to the primary trauma [7,8]. The key compound involved in the growth of the injured area is the excitatory neurotransmitter of the central nervous system (CNS)—glutamate [10,11]. Shortly after the trauma, its local concentration in the traumatized brain can increase up to 50 times that in a normal state [12]. Glutamate toxicity is determined by hyperactivation of ionotropic NMDA receptors leading to biphasic acceleration of the cytoplasmic concentration of Ca^2+^ ([Ca^2+^]_i_) [13,14]. Under physiological conditions, the Ca^2+^ concentration inside the cell is 10^5^ times less than outside [15]. This tremendous gradient is sustained by a sophisticated energy consuming buffering system that comprises two main constituents: the mitochondrial Ca^2+^ uniporter (MCU) and plasmalemmal buffering system [15]. Mitochondria use an electrochemical gradient to transport Ca^2+^ ions from the cytoplasm through the MCU. The plasmalemmal buffering system consists of Ca^2+^ ATP-ase and Na^+^/Ca^2+^ uniporter, which uses the Na transmembrane gradient created by Na^+^/K^+^ ATP-ase to expel Ca^2+^ ions into the intercellular space [13,16,17]. If the inward Ca^2+^ current surpasses the capacity of the buffering systems, the mitochondrial Ca^2+^ concentration reaches a certain level that triggers mitochondrial inner membrane permeabilization for molecules less than 1,5 kDa [18]. This process is followed by abrupt inner membrane depolarization and a sharp increase of [Ca^2+^]_i_ called delayed Ca^2+^ deregulation (DCD)—the hallmark of inevitable cellular death [19,20] with a necrotic or apoptotic phenotype depending on the energy supply availability [1,21]. Thus, substances that can have a positive effect on intracellular Ca^2+^ buffer systems or the cell energy supply can potentially reduce cell death from TBI.

Due to the diverse and complex nature of TBI-related adverse events, the efficiency of its therapy remains rather low [22,23]. Therefore, the search for therapeutic substances that may oppose the pathology progression without side effects is an urgent task in the field of neurotraumatology. The endogenous tripeptide Pro-Gly-Pro (PGP) is produced through hydrolysis of either collagen or elastin by prolyl endopeptidases and oligopeptidases of activated immune cells (neutrophils, microglia cells, and astrocytes) [24,25]. The commercially available nootropic and neuroprotective heptapeptide Semax contains PGP at its C terminus. Semax also can be used as an exogenous source of PGP that is released upon cleavage by endogenous peptidases [26]. In various in vivo models of total and local rat cerebral ischemia, it has been repeatedly shown that under the influence of PGP, the levels of expression of genes encoding trophic and growth factors (BDNF, VEGF, TGF-β, etc.), as well as their receptors (Trk, VEGFR, etc.) are increased significantly; this fact indicates its neuroprotective effect [27]. Additionally, PGP downregulates the expression of complexin-2 protein, which is involved in exocytosis from synaptic ends [28]. This can reduce the excitotoxic release of glutamate during ischemia and promote neuroregeneration. PGP was shown to bind neuronal membranes, but this interaction is fragile and is eliminated by the addition of selective NMDA antagonists, GABA (γ-aminobutyric acid), endocannabinoid receptors, and dopamine receptors. There may be allosteric binding and regulation of these receptors, but there is no exact evidence for this [29].

We hypothesized that PGP has a neuroprotective effect due to the wide spectrum of protective actions and its lack of toxic effects. In this work, we studied the effects of PGP on the survival and changes in [Ca^2+^]_i_ and ΔΨm of neurons in primary neonatal rat cortical culture after mechanical injury. We also assessed the neuroregenerative potential of the tripeptide in the in vitro experimental model of neurotrauma (scratch test) that we had previously improved and adapted for primary neuroglial culture [6].

## 2. Results

### 2.1. The Effect of PGP Treatment on the Survival and Regeneration Potential of Neuro-Glial Culture Cells after Mechanical Trauma

To study the neuroregenerative potential of the peptide, we used our recently tested and improved experimental model of mechanical neurotrauma in vitro (the scratch test). The consequences of mechanical damage to the primary culture of cortical cells of neonatal rats have been studied and described in detail [6]. In brief, the essence of the experimental model is the mechanical disintegration of the neuronal network formed in the primary neuroglial culture at 4 DIV by applying a scratch. Such damage leads to the immediate death of all cells in the scratch area and to a subsequent reduction in their number in nearby areas (Figure 1).

In this study, disintegration of the neuronal network also led to the immediate death of the damaged cells followed by a gradual decrease in the viability of the surrounding cells during the next two weeks as compared to undamaged cultures (Figure 2A,B). The MTT test showed that mechanical trauma to the cell culture led to a 20% reduction in its survival rate compared to the control (*p* < 0.001). Application of the peptide significantly increased cell survival (Figure 2A). The metabolic activity of injured cultures treated with PGP increased by 16% (*p* < 0.05). In addition, the ratio of live/dead cells increased in areas close to the scratch (Figure 2B). With time, the number of living cells in the injured area increased, which was the evidence for their migration (Figure 2C and Figure 3C).

Immunocytochemical staining showed the neuroregenerative effect of the peptide was already evident on the 3–7th day after the trauma (Figure 3B,D), which manifested as a higher number of living cells found in the scratched area of PGP-treated cultures (Figure 3D) compared to the untreated control (Figure 3C). The addition of PGP to the wells not only increased cell survival in areas close to the scratch, but also decreased astrocyte hypertrophy (Figure 3B). We also registered an accelerated overgrowth of the scratched area with neuronal processes in damaged cultures treated with PGP (Figure 3D). Figure 2C–D show the semiquantitative representation of this process. The effect of the peptide on the process of astrogliosis is shown in Figure 3E–G. On the 3rd day after the scratch, i.e., in the acute period of injury, a 2.5 times decrease in total GFAP fluorescence (Figure 3E) and a 3.2 times increase in total β-III tubulin fluorescence (Figure 3F) were noted. In general, the β-III tubulin/GFAP ratio in the injured and PGP-treated neuroglial culture was significantly higher than in the injured and untreated group (Figure 3G), indicating the neuroregenerative potential of the peptide. On the 7th day after the scratch, the difference between the groups was also significant, although the effect of the peptide slightly decreased, which affected the β-III tubulin/GFAP ratio.

Thus, mechanical trauma causes delayed cell death in the primary neonatal rat cortical cells. Disintegration of the cortical network decreased the viability of the cells in culture as was supported by both MTT tests and live/dead ratio counting. In turn, the addition of PGP to the injured cell culture positively influenced both of these parameters. Additionally, the peptide promoted the migration of cells and accelerated the growth of processes toward the injury area.

### 2.2. Effect of PGP on the BDNF and NSE Levels in Neuroglial Cell Culture after Mechanical Injury

Brain-derived neurotrophic factor (BDNF) is a neurotrophin secreted by CNS neurons and astrocytes that is involved in neuronal survival and synaptic plasticity [30]. Determination of the ratio showed that the intracellular and extracellular levels (Ratio_Intra/Extra_) of BDNF had significantly reduced 1 day after scratch application (Figure 4A), primarily due to a sharp decrease in its intracellular content (Figure 4C). The presence of PGP (30 µM) in the culture significantly increased the BDNF Ratio_Intra/Extra_ (Figure 4A), mainly due to an increase in BDNF expression in cells (Figure 4C). Subsequently, the BDNF level in the presence of the peptide remained slightly elevated through 3 days post-scratch with no pronounced changes at 7 days. Its extracellular levels in the presence of the peptide did not differ from the control (Figure 4E).

The NSE Ratio_Intra/Extra_ also decreased significantly (Figure 4B), but in this case due to an increase in its extracellular content (Figure 4F). This may be due to the disruption of the cell membrane and the release of protein into the extracellular environment. A high level of NSE in the blood serum correlates with the severity of TBI [31]. In the presence of PGP, the release of NSE into the extracellular medium decreased after just 1 day and remained at a low level throughout the experiment (Figure 4F). However, the peptide did not have a significant effect on its intracellular content (Figure 4D).

### 2.3. The Effect of Mechanical Injury on the Parameters of Acute Changes of [Ca^2+^]_i_ and ΔΨm

It is known that mechanical trauma disrupts Ca^2+^ homeostasis in affected nervous tissue and triggers abnormal neuronal depolarization during neuro-trauma [6,32]. Therefore, we wanted to test the ability of the PGP tripeptide to influence Ca^2+^ homeostasis. In this series of experiments, changes in [Ca^2+^]_i_ caused by directly inflicted trauma were measured on the 4 DIV. Temporary or DCD-like changes in [Ca^2+^]_i_ were registered in response to trauma (Figure 5A). The strongest [Ca^2+^]_i_ fluctuations were detected in the cells closest to the area of the scratch (Figure 5B,C). Some fluorescent responses of mechanically damaged cells had a high amplitude with linearly decreasing dynamics (Figure 5A, group of neurons C’). Most of the remotely-located cells had a low amplitude and rapidly passing fluorescence response (Figure 5A, group of neurons A’). At the same time, both in the near and far distance from the scratch there was a significant number of cells that did not react to the injury, and they were not taken into the account (Figure 5B,C). Among cells with a high amplitude fluorescent response, there were those that responded to trauma by the development of a two-phase Ca^2+^ influx (Figure 5A, neurons three to five), similar in dynamics to DCD; there were about 25% of such cells in both groups (Figure 5A,B). We measured [Ca^2+^]_i_ changes only in cells that were not directly affected by injury but still developed two-phase Ca^2+^ influx (Figure 5A, group of neurons B’). We counted a number of such cells with a lag phase of secondary influx higher than 100 sec (Figure 6B) and measured the average longitude of the lag period of the secondary influx (Figure 6A). The lag period duration represents the metabolic capacity of the cell to resist mechanically induced Ca^2+^ influx. PGP-treated cultures had a higher fraction of cells with late onset of secondary Ca^2+^ influx than in untreated cells, respectively 47% vs. 19% (Figure 6D,E). Moreover, the longitude of the lag phase in cells from treated cultures was significantly higher (*p* < 0.05) (Figure 6C).

Therefore, we can conclude that pre-treatment of neuro-glial cultures with PGP tripeptide improves resistance of neurons to potentially cytotoxic fluctuations in [Ca^2+^]_i_ caused by the application of mechanical injury.

### 2.4. The Effect of Mechanical Injury on the Delayed Changes of [Ca^2+^]_i_ and ΔΨm

To further study the influence of the PGP tripeptide on the functional status of the intracellular Ca^2+^ homeostasis of neurons, artificial conditions for glutamate excitotoxicity were created. On day 3 post-injury, the neuroglial cultures were subjected to a subtoxic dose of glutamate—33 µM. Changes in [Ca^2+^]_i_ and ΔΨm in cultured neurons in response to the external application of glutamate characterize the functional state of the cells and the ability of their mitochondria to synthesize ATP and take up the excess Ca^2+^ entering the cytosol via Glu-activated ionotropic receptors.

During the experiment, three groups of cultures were formed: “Control” group that did not receive either a traumatic impact or PGP administration, an “Injury” group that was injured on 4 DIV, and an “Injury + PGP” group that received PGP treatment daily after injury—during the 4–10 DIV period. Only cells near the scratch (no farther than 50 mM) were evaluated in the last two groups.

Two qualitative parameters that represent the state of cellular Ca^2+^ homeostasis were measured: the time of delayed Ca^2+^ dysregulation development (lag-period DCD) and the potency of low cytoplasmic Ca^2+^ content recovery upon excitotoxic stimulus abrogation.

The lag period of DCD represents the ability of the cell to resist the excitatory stimulus of glutamate receptors overstimulation. Upon the application of a toxic glutamate dose, a high amount of calcium enters the cytoplasm though the NMDA receptors. For some period, neurons can either expel it out of the cell or load it into the mitochondria. The advent of DCD happens when the mitochondrial capacity runs out. Therefore, the longitude of the DCD lag period depends on the mitochondrial calcium capacity and expelling system effectiveness. Next, the potency of Ca^2+^-recovery upon removal of the glutamate and Ca^2+^ from the experimental solution represents the ability of the cell to restore the mitochondrial functions of the Ca^2+^ uptake from the cytoplasm. Both the Ca^2+^ capacity of the mitochondria and the ability to restore the buffering functions after the DCD depends on the parameters of the mitochondrial permeability transition pore formation process. Injured neurons display a significantly faster DCD attack than intact ones, and the administration of PGP helped to prolong this period (Figure 7(A1,A2)).

The same tendency with an even greater difference between the groups was displayed by a parameter of the [Ca^2+^]_i_ recovery, which was counted as $\left(\frac{a-b}{a}\right)*100\%$, where *a*—is the maximal [Ca^2+^]_i_ during glutamate application and *b*—is the minimal [Ca^2+^]_i_ during the EGTA application (Figure 7(B1,B2)). This array of facts may suggest that the disturbances in calcium homeostasis caused by mechanical injury delivered to the culture were restored more effectively by PGP application.

Moreover, we acquired congruent results during the registration of the Rhod123 fluorescent signal that represents changes in Ψm (Figure 7(C1,C2)). Mitochondria are one of the main Ca^2+^ storage compartments in neurons, and their function strongly depends on their ability to preserve the electrochemical potential of the inner membrane. We observed that inner mitochondrial membranes were depolarized much more strongly under conditions of glutamate excitotoxicity in cells from injured cultures than in intact cultures. PGP administration changed the dynamics of mitochondrial depolarization and repolarization in a protective manner (Figure 7(C1,C2)).

## 3. Discussion

Traumatic brain injury (TBI) and accompanying ischemia/hypoxia of brain tissue initiate an excitotoxic cascade [1] and free radical injury followed by inflammation, producing injury in neurons, glial cells and white matter [21]. Therefore, the search for therapeutic substances that may oppose the pathology progression without side effects is an urgent task in the field of neurotrauma. It is known that proline- and glycine-containing oligopeptides, including PGP, are formed as fragments of the protein matrix of the body’s connective tissue; this process is associated with collagen synthesis in fibroblasts and its catabolism [33,34]. In certain quantities, the peptide PGP (350 pg/mL) and its acetylated form AcPGP (6.3 pg/mL) are detected in the serum of healthy individuals [35]. It is known that PGP can cross the blood–brain barrier, and intranasal administration of the peptide has been shown to be optimal for delivering glyproline molecules to the central nervous system [36]. In this study, our primary focus was on investigating the neuroregenerative properties of PGP, as well as its ability to protect neuro-glial cells from secondary processes that occur immediately following mechanical damage of the cell cultures, leading to delayed neuronal death. In the context of traumatic brain injury therapy, such treatment may prevent the sequential loss of neurons and the development of post-traumatic syndrome. We showed that PGP increases cellular viability and accelerates the healing of the neurotrauma in vitro, carried out according to a previously developed and published protocol [6]. Its neuroprotective and neuroregenerative potential may be due to several effects of the peptide, which are collectively illustrated in Figure 8.

Firstly, brain damage in TBI leads to an increase in the BBB permeability and the release of protein neuromarkers NSE and S100b into the blood [37]. In our study, the level of extracellular NSE decreased against the background of the action of PGP, which indicates a decrease in the number of damaged cells in culture one and three days after injury. In addition, TBI may influence neurotrophic signaling via the BDNF expression change [38,39]. Brain-derived neurotrophic factor (BDNF) is a neurotrophin secreted by CNS neurons and astrocytes that is involved in neuronal survival and synaptic plasticity [40]. It is known that after TBI, serum BDNF is acutely decreased, correlating with injury severity and adverse outcomes in children with severe TBI [37], and therapies that increase brain BDNF expression show promise for cognitive recovery [41]. This study has shown that 24 h (1 day) after mechanical injury of the cortical culture, there was a dramatic decline in the content of BDNF both inside the cells and in the extracellular fluid, which may be associated with a decrease in its expression in cells. When PGP was added to the mechanically damaged cell culture, the level of intracellular BDNF significantly increased. Three and seven days after the scratch test, the content of BDNF in both groups tended to increase. Perhaps this was due to the repair processes that start from the onset of brain injury. The absence of a difference between the control and the experimental group on day 7 may be attributed to the natural healing of the primary neuroglial culture, which we showed earlier [6].

Secondly, it is known that neurons can express chemokine receptors, which allows their targeted migration during normal brain development [42,43]. It is known that PGP has chemoattractant properties in relation to several cells [24], although the effect is not as pronounced as its acetylated form—Acetyl-Pro-Gly-Pro (AcPGP) [44]. Therefore, PGP addition to the cultural media could cause an increased motility and migration of neurons and glial cells in the direction of the scratch. As immunocytochemical staining of β-III tubulin and Hoechst 33342 showed, the number of neurons in mechanically damaged and neighboring scratch areas significantly increased upon the application of PGP. Patel and Snelgrove anticipated that PGP essentially functions as a matrikine, which is generated during infection or injury and acts to shape the ensuing inflammatory and repair processes. As a fragment of the ECM that accumulates at the epicenter of the action, PGP is perfectly positioned to focus neutrophils to exactly where they are required, as well as initiate proximal repair responses [45]. Acetylated PGP was shown to exert its chemoattractant action on neutrophils via CXCR2 [46]. It is known that both astrocytes [47,48] and neurons [49] are capable of upregulating CXCR2 on their surface under inflammatory conditions. Therefore, we can speculate that PGP-driven migration of neurons into the scratched area involves signaling via CXCR2.

This may accelerate the healing of neurotrauma. However, it is likely that its ability to transform into AcPGP and persist in vivo for a long time in the acetylated form may lead to pathological remodeling of epithelial tissue. There is evidence that the long-term presence of AcPGP may lead to pathologies observed in chronic lung diseases [46]. Despite this, it is clear that both PGP and AcPGP molecules are important players in maintaining tissue architecture homeostasis.

Thirdly, other than direct impact mechanical damage to the cells, traumatic brain injury can cause additional neuronal cell death by triggering an inflammatory process involving astrocytes and microglia [50,51]. A primary culture of astrocytes from the cortex of newborn rats after mechanical damage exhibited behavior reminiscent of the process of astrogliosis—hyperplasticity, with morphological changes and increased GFAP content [6,52]. As immunocytochemical staining of GFAP showed, the reactivity of astrocytes significantly decreased with the use of PGP. Treatment of the injured neuroglial culture with the peptide contributed to a reduction in astroglial scars; the astrocyte bodies were small in size, and their processes were much thinner. The anti-inflammatory effects of PGP were already pointed out by our group and others. For instance, PGP had a protective effect on the gastric mucosa in a model of ulcerogenesis [53] by downregulating the expression of the pro-inflammatory cytokine Gro/Cinc-1 in gastric epithelium cells [54]. In another study, PGP reduced the production of the IL-1β by ln. gastricus caudalis cells, and its level in serum, thus limiting local inflammatory processes and reducing the formation of stress-induced ulcers [55]. The anti-inflammatory effect of PGP was observed in vitro on mast cells activated with tetracosactide (Synacthen Depo^®^). It preserved the cell morphology and inhibited histamine secretion [56]. In addition, PGP exerted anticoagulant and hypoglycemic effects in patients with hypercholesterolemia [26]. This may have an additional therapeutic effect after suffering a TBI.

And finally, a decreased activity of NMDAR due to PGP application could increase the viability of cells in neuro-glial culture. We previously showed that the entry of Ca^2+^ into cells during mechanical damage to the primary neuroglial culture occurred predominantly through the NMDAR [6]. MK801, an inhibitor of NMDAR, prevented the acute increase of the [Ca^2+^]_i_ in 99% of the neurons. In addition, pathological changes in calcium homeostasis persisted in the primary cortical culture for a week after the injury [6]. In our experiments, a single application of PGP (100 μM) prevented an acute increase in intracellular calcium [Ca^2+^]_i_ and a sharp drop in mitochondrial potential [ΔΨm] in a mechanically damaged neuronal culture, which was accompanied by a decrease in cell death. Now, we have demonstrated that the mechanisms of the neuroprotective and neuroregenerative action of PGP also lie in its ability to reduce the long-term consequences of mechanical damage to neuroglial cells, a specifically delayed disruption of calcium homeostasis and neuronal death. In experiments studying calcium homeostasis a week after injury, PGP prevented the glutamate-induced changes in [Ca^2+^]_i_ and [ΔΨm] in a damaged neuronal culture. It is known that TBI, including mechanical brain injury, is accompanied by glutamate excitotoxicity [11]. Our earlier studies also demonstrated the protective effect of PGP on cerebellar granule neurons [57] and cortical neurons [58] under the influence of toxic doses of glutamate (100 μM). In the latter case, among other studied drugs, PGP had the highest potential to delay excitotoxic glutamate-induced Ca^2+^ deregulation in neurons.

It was shown before that PGP stimulates the expression of neurotrophic factors, in particular VEGF [27]. VEGF, in its turn, increases the expression level of the Bcl-xl gene [59], a protein that directly blocks permeabilization of the outer mitochondrial membrane [60] and helps to alleviate symptoms induced by preliminary mechanical injury [61]. The assumption that mechanical trauma downregulates and PGP upregulates the expression of Bcl-2 family proteins and that it shifts the sustainability of [Ca^2+^]_i_ and mitochondrial potential homeostasis might explain our results. Other mechanisms of PGP-attributed neuroprotection might be its positive influence on mitochondrial metabolic status, as it was shown to enhance the survival of pheochromocytoma cells under conditions of oxidative stress [62].

However, this notion does not explain how a one-time application of PGP shortly before the trauma can influence the rapid changes in [Ca^2+^]_i_ inflicted by injury. This implies that PGP could have more than one mechanism of action. It is known that the expression and duration of growth factors, which PGP stimulates in vivo [27], have a longer kinetics [63]. Therefore, we assume that PGP may affect membrane-mediated intracellular cascades associated with the regulation of Ca^2+^ homeostasis systems. There is evidence that PGP can reversibly bind to NMDA receptors [29]. Perhaps such a binding could directly or indirectly affect the kinetics of NMDA NR2B subtype receptor action upon their activation by glutamate. Thus, PGP is an important molecule in the process of organizing the local restorative response.

## 4. Materials and Methods

### 4.1. Primary Rat Cortical Cultures

Primary rat cortical cultures were prepared from the cortex of 1- or 2-day-old Wistar rats. The rats were anesthetized, decapitated, and the cortex was removed and separated from the meninges. The extracted tissues were washed in a Ca^2+^- and Mg^2+^-free Hanks solution, crushed and placed for 15 min in a papain solution at 36 °C, washed with standard Hanks solution with phenol red and MEM (Minimal Essential Medium) culture medium, and dispersed in fresh MEM. A homogeneous suspension was precipitated two times for 5 min in a centrifuge (Jouan BR4, Lucé, France) at 1000 rpm. The precipitated cells were resuspended to a concentration of 10^6^ cells/mL in neurobasal medium (NBM) with the addition of Supplement B-27, GlutaMax, and penicillin/streptomycin. The cells were seeded on either glass coverslips attached to the wells of 35 mm plastic Petri dishes (“MatTek”, Ashland, MA, USA) or 48-well plates (Corning costar) at a concentration of 40,000 cells per cm^2^. The glasses and plates were pre-coated with 1 mg/mL of polyethyleneimine (PEI) for 30 min. The cells were kept in an incubator at 37 °C, 95% air + 5% CO_2_, and a relative humidity of 100%. Arabinosine monocytoside (AraC, 5 μM) was added to the medium on the 2nd or 3rd day to prevent glial cell proliferation. Every three days, the cells were fed by replacing 1/3 of the old medium with new medium.

### 4.2. Scratch Test and Migration Rate

The cells were cultured in 48-well plates to perform the previously described scratch test [6]. Briefly, on a third day in culture, the formed neuroglial network was scratched with a 200 µL disposable pipette tip (Eppendorf, Hamburg, Germany). Then, 30 μM PGP (IMG RAS, Moscow, Russia) was added immediately after scratching to the experimental wells and every 24 h thereafter until the described assays. Intact neonatal rat cortical cultures and untreated damaged cultures served as controls. On days 6, 10 and 13 of the culture, the number of cells in the scratch area was calculated. For this, the cells were stained with a calcein-AM/ethidium homodimer-1 (EthD-1) Live/Dead Viability/Cytotoxicity Kit for mammalian cells, then rinsed with PBS thrice and observed by an EVOS FL Auto imaging system to record the green live cells and red dead cells. The cell migration rate as described above was used to detect the speed of healing of the neuroglial culture after the scratch.

### 4.3. Evaluation of Neuronal Viability

The effect of PGP on the cell viability in disintegrated cortical cultures was evaluated using an MTT [3-(4,5-dimethylthiazol-2-yl)-2,5-diphenyltetrazolium bromide] assay and morphological analysis.

The MTT assay was performed on days 10–13 of the culture in 48-well plates. The MTT solution (0.5 mg/mL in culture medium) was added to the wells for 1 h in a CO_2_-incubator at 37 °C. The culture medium was then removed and DMSO (Thermo Fisher, Waltham, MA, USA) was added to dissolve the formazan, a reduced tetrazolium salt. The optical density of the solution was measured using a ClarioStars multimodal plate reader (BMG Labtech, Offenburg, Germany) at a wavelength of 520 nm and a reference wavelength of 690 nm. The obtained data were normalized by taking the absorption of formazan in control cultures as 100%.

Morphological analysis of the number of live and dead cells was performed as previously described [6]. Briefly, the cells were loaded with fluorescent dyes (15 min, 37 °C) using 1:1000 dilutions of DMSO stock solutions in a saline buffer. Syto-13 (1 μM, ex 485/em 530 nm) was employed to determine live cells. Necrotic cells were identified by staining with ethidium homodimer (EthD-1) (2 µM, ex 565/em 610 nm). The nuclei both live and dead cells were stained with Hoechst 33342 (1 μM, ex 343/em 483). To obtain images (20× magnification) and count the cells, the EVOS FL automated imaging system was used. Neuronal viability was assessed with the ratio of the living to the dead cell numbers after 1 h at 3, 7, 10, and 14 days after injury (4, 7, 11, 14, and 18 DIV, respectively). The data obtained were normalized relative to the mean values in the control wells. The efficiency of injured cultures regeneration was defined as the number of neurons that moved to the damaged area.

### 4.4. Immunofluorescence Staining of Neuroglial Culture on Markers GFAP and Beta-3 Tubulin

The cell phenotype in the cultures was determined by immunofluorescence staining. The cultures were fixed with paraformaldehyde (4%, 15 min) and permeabilized with ice-cold methanol for 15 min. Non-specific binding was blocked with 2% BSA in PBS (20 min, room temperature). The cells were incubated overnight at 4 °C with anti-β-III tubulin (2G10) mouse monoclonal antibodies (Thermo Fisher, #MA1-118; dilution of 1:100) and anti-GFAP chicken polyclonal antibodies (Thermo Fisher, #PA1-10004; dilution of 1:1000). Then, the cells were washed with PBS and incubated (30 min, room temperature, dilution 1:100) with fluorescently labeled anti-mouse IgG (Thermo Fisher, #A32727; Alexa Fluor Plus 555) and anti-chicken IgY (Sigma–Aldrich, St. Louis, MO, USA, #SAB4600031; CFTM 488A). For counting the cells after immunofluorescent staining, their nuclei were labeled with Hoechst 33342 (1 μM, 10 min, room temperature).

Images of immunofluorescently labeled cells were obtained employing an LSM 880 laser scanning confocal microscope equipped with an AiryScan module and GaAsP detector (Carl Zeiss, Jena, Germany). Tile scans (2 × 2; image size was 512 × 512 pixels) were collected with a Plan-Apochromat 40×/1.2 mm Corr DIC M27 multi-immersion objective. Laser lines at 405 nm, 488 nm and 561 nm were used to excite the fluorescence of Hoechst 33342, and secondary anti-chicken IgY and anti-mouse IgG antibodies, respectively. Fluorescence emission was collected at 410–507 nm for Hoechst 33342, at 495–530 nm for anti-chicken IgY, and 567–675 nm for anti-mouse IgG.

### 4.5. Determination of Neuromarkers BDNF and NSE

The mixed cortical neurons cultures were injured on the 4th day in culture (4th DIV). The number of cells in each well was 5 × 10^5^. Extra- and intracellular concentrations of BDNF (SEA011Ra, Cloud-Clone Corp., Katy, TX, USA) and NSE (SEA537Ra Cloud-Clone Corp., Katy, TX, USA) were measured with ELISA kits 1, 3 and 7 days after the scratch was applied. For release of intracellular proteins, 0.5 mL of lysis buffer (IS007, Cloude-Clone Corp., Katy, TX, USA) was used per well. The extracellular concentration of the proteins was determined in conditioned medium. Four 48-well plates with non-sister cultures were used. The number of wells of culture plates for each point is 5–6 (n = 5–6). The results were analyzed using two-way ANOVA with a subsequent Tukey’s multiple comparisons test and presented in the format mean with SD, as well as the ratio of intracellular to extracellular protein content. The results were considered statistically significant with *p* < 0.05.

### 4.6. Application of Mechanical Injury during Calcium Imaging

Mechanical injury was applied to a 4 DIV neuroglial culture loaded with Fura-FF and Rhd123 during the registration of fluorescent signals as described previously [6]. Briefly, to measure changes in [Ca^2+^]_i_ and ΔΨm immediately after the trauma (the acute effect), we applied a single scratch trauma across the neuroglial culture with an insulin injection needle attached to the tip of a rod-like metal holder connected to a micromanipulator (Narishige, Tokyo, Japan). The cells were preincubated with PGP (30 μM) for 1 h. The images were recorded every 3 s during the first 3 min following scratching and then every 30 s for 12 more minutes. In other experiments, the neuroglial cultures (4 DIV) were scratched across the diameter of the well bottom by a disposable sterile tip (Eppendorf, 200 µL). The damaged neuroglial culture was treated with PGP for 3 or 6 days (once daily). Then, we evaluated the delayed effect of mechanical damage on the cell viability and examined the long-term changes of [Ca^2+^]_i_ or ΔΨm on the 3rd (7 DIV) and 7th (11 DIV) days after scratching, as detailed in the next section.

### 4.7. Measurements of Changes of Cytosolic Free Ca^2+^ Concentration ([Ca^2+^]_i_) and Mitochondrial Potential (ΔΨm)

Fluorescence microscopy measurements of [Ca^2+^]_i_ and ΔΨm were performed on cultured neurons at 8–11 days in vitro (DIV) with an imaging system consisting of an inverted microscope Olympus-IX71, excitation/emission dual filter-wheel controller (Lambda 10–2, Sutter Instruments Co., Novato, CA, USA), and CCD camera CoolSnap HQ2 (Photometrics, Tucson, AZ, USA). Measurements were performed using a 20× objective. Images were acquired and analyzed using MetaFluor 6.3 and MetaFluor Analyst programs (Molecular Devices, San Jose, CA, USA). To monitor [Ca^2+^]_i_ and ΔΨm, the cells were loaded with two fluorescent indicators, Ca^2+^-sensitive Fura-FF (4 µM, 45 min; in 0.02% Pluronic F-127) and a potential-sensitive cationic probe Rh123 (6.6 µM, 15 min) at 37 °C in culture medium, respectively, followed by thorough rinsing with N-buffer (mM): 135 NaCl, 5 KCl, 2 CaCl_2_, 1 MgCl_2_, 20 HEPES, 5 D-glucose, pH 7.4. Pluronic F-127 (0.02%) was used to facilitate the loading.

The cells were subjected to sequential incubation in the following solutions: N-buffer (5 min), Glu (15 min), EGTA (15 min), FCCP (5 min), and ionomycin (5 min). Glu (33 μM) was added in Mg^2+^-free saline buffer containing glycine (10 μM). To wash out the Glu, a nominally Ca^2+^-free buffer containing 0.1 mM EGTA and 2 mM Mg^2+^ was used. The measurements were performed as described in [BMM] at the temperature 27–29 °C. The images were recorded every 30 s under excitation 340 nm and 380 nm for Fura-FF and 485 nm for Rh123. The emission of both dyes was recorded at 525 nm.

Relative changes of [Ca^2+^]_i_ are presented according to Equation (1):(1)$$[{\mathrm{Ca}}^{2+}{]}_{i}=\frac{R-R_{\min}}{R_{\max}-R_{\min}}$$

To determine the maximal R_max_, a Ca^2+^-specific ionophore (ionomycin 2 µM, in the presence of 5 mM Ca^2+^ in the buffer) was applied at the very end of the experiment.

The latent period of the Glu-induced delayed calcium deregulation (DCD), the percentage of cells that underwent DCD, and the percentage of cells that recovered low [Ca^2+^]_i_ after Glu washout were determined as described earlier [6]. The parameters characterizing the functional state of Ca homeostasis were the amount of Ca that accumulated in the neuronal cytoplasm during the DCD, the rate of Ca extrusion after the ablation of excitotoxic conditions, the extent of mitochondrial depolarization during the DCD and the propensity of mitochondrial potential recovery.

Data were acquired using MetaFluor software the version 1.0.93 as images of the entire observed area and as a Microsoft Excel table for individually selected areas. The recorded images were processed in MetaFluor Analyst software (Molecular Devices, San Jose, CA, USA).

### 4.8. Reagents

The cell culture supplies were obtained from Invitrogen (Thermo Fisher Scientific, Waltham, MA, USA). Fluorescence microscopy dyes were acquired from Molecular Probes (Thermo Fisher Scientific, Waltham, MA, USA). All other reagents were purchased from Sigma–Aldrich (Merck, St. Louis, MO, USA). We also used several conventional media, optimized for the particular types of experiments.

### 4.9. Statistical Analysis

Statistical data analysis was performed with aid of GraphPad Prizm the version 8.1.0 (GraphPad Software, San Diego, CA, USA). All experiments were performed in duplicate at least 3 times on non-sister cell cultures. Normality of distributions was assessed using the Shapiro–Wilk test. Data from the MTT test were analyzed using one-way ANOVA with the Holm–Šidak correction; if the data distribution differed from the normal distribution, the Kruskal–Wallis test was used with a subsequent Dunn’s multiple comparisons test. Quantitative data from protein content assays were analyzed using two-way ANOVA with subsequent Tukey’s multiple comparisons test. Calcium imaging and ΔΨm data were analyzed using the Mann–Whitney test (for 2 groups) or the Kruskal–Wallis test (for more than 2 groups), adjusted for multiple comparisons (Dunn’s multiple comparisons test). At least three independent experiments were conducted in each series. Data are presented as means and standard deviations or medians and interquartile ranges. Differences were considered statistically significant at a confidence level of *p* ≤ 0.05. A more detailed analysis of the results should be found in the specific method’s description.

## 5. Conclusions

TBI can initiate a very complex disease of the CNS, starting with the primary pathology of the inciting trauma and subsequent inflammatory neuronal tissue response. We have shown that the peptide PGP has an anti-inflammatory effect by reducing astrocyte hyperactivation and NSE expression. Moreover, the application of PGP right at the beginning of the pathological process can increase the viability and migration potential of neurons under conditions of mechanical injury, as evidenced by the scratch test and increased synthesis of BDNF in cortical culture. This may accelerate the healing of neurotrauma. However, there may be some concerns, since there is evidence that long-lasting PGP persistence can drive the pathologies observed in chronic lung diseases. Whilst, as PGP may promote repair of neuroglial culture in vitro, probably its ability to transform under in vivo conditions into an acetylated form (AcPGP) can result in pathological remodeling of the tissue architecture. Despite this, PGP has different mechanisms of neuroprotective action that likely involve direct interaction with NMDA-receptors, because it positively affects Ca^2+^ homeostasis in neurons and mitochondrial potential. Mitochondrial Ca^2+^ overload and a fall in ΔΨm can cause a delayed neuronal mortality and thus play a key role in the development of the post-traumatic syndrome. Prevention of the prolonged mitochondrial depolarization and calcium dysregulation of neurons with PGP may be a promising therapeutic approach to increase neuronal survival upon traumatic brain injury.

## Figures and Tables

**Figure 1 ijms-25-10886-f001:**
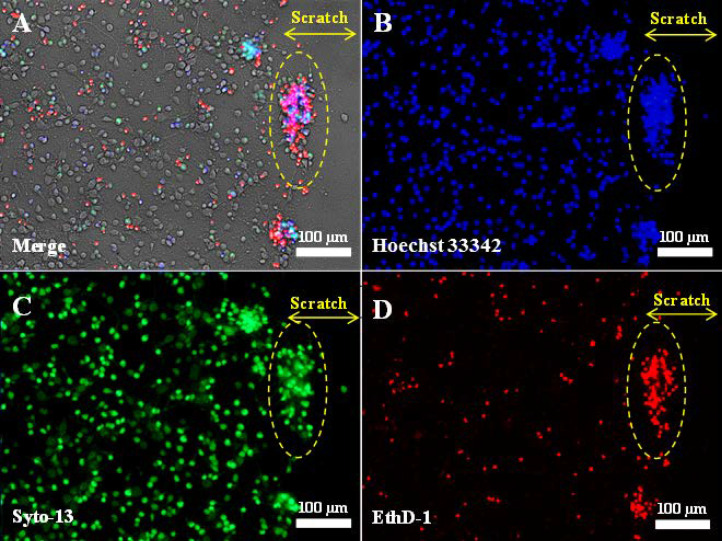
Fluorescent images of primary neuroglial culture after mechanical injury (7 days in vitro, 3 days post-scratch application): phase contrast (**A**); nuclei stained with Hoechst 33342 (**B**); live cells stained with Syto-13 (**C**); dead cells stained with EthD-1 (**D**). The scratch was applied using a disposable pipette tip (Eppendorf, 200 μL) on the 4th day after cell seeding, when the cells had formed a neuronal network. Clusters of live and dead cells at the edge of the scratch are outlined by a dashed oval. The yellow double arrow indicates the width of the scratch. No live cells are present in the scratch zone.

**Figure 2 ijms-25-10886-f002:**
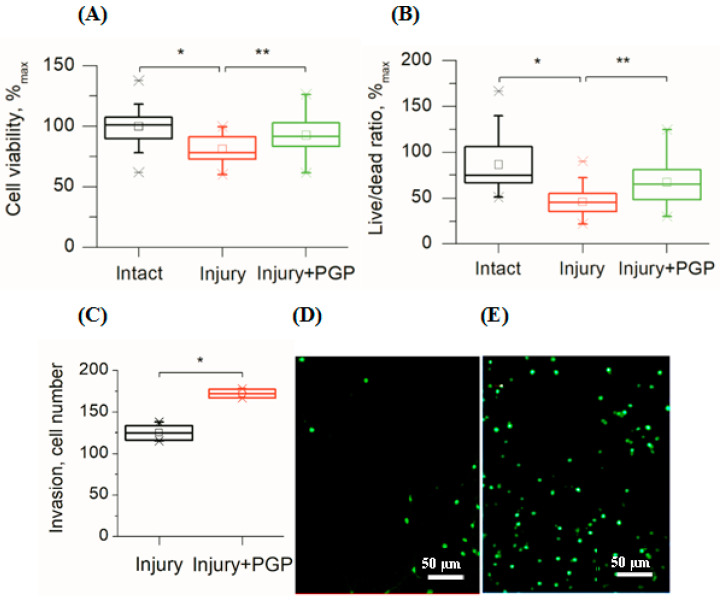
Assessment of cellular viability of mechanically injured rat primary cortical cell culture under PGP treatment (the 7th day after the injury). (**A**) MTT test; (**B**) Live/Dead cell ratio (Syto-13/EthD-1) in areas close to the scratch; (**C**) the number of living cells invaded per field of view in the scratch area; (**D**) living cells (Syto-13) in the injured area in untreated control culture and (**E**) after treatment with PGP (30 µM). Data are presented as mean ± SD, statistically significant differences were detected using a one-way analysis of variance adjusted for multiple comparisons (Holm–Sidak’s multiple comparisons test). * *p* < 0.05; ** *p* < 0.01.

**Figure 3 ijms-25-10886-f003:**
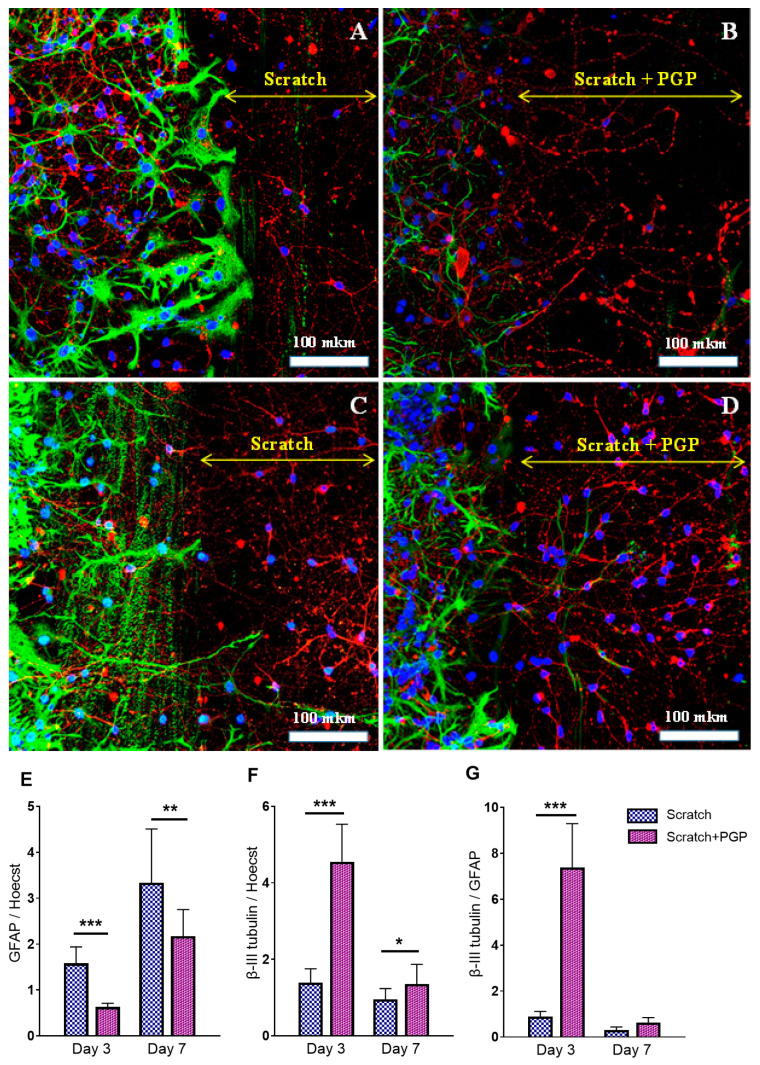
Immunocytochemical staining of neuroglial culture for GFAP (green) and β-III tubulin (red). Nuclei were stained with Hoechst 33342. Fluorescent images of the cell culture were obtained at day 3 after the scratch, i.e., 7 DIV (**A**,**B**), and at day 7 after the scratch, i.e., 10 DIV (**C**,**D**). The peptide PGP was added within 3 days (**B**) or 6 days after the scratch (**D**). Addition of the peptide reduced astrogliosis and increased migration of neurons into the damaged area; many nuclei were observed at the scratch border. The diagram below shows the semiquantitative representation of this process. The ratio of GFAP (**E**) and β-III tubulin (**F**) fluorescence to Hoechst 33342 fluorescence. The ratio of β-III tubulin fluorescence to GFAP fluorescence (**G**). * *p* < 0.05; ** *p* < 0.01; *** *p* < 0.005. There was a decrease in total GFAP fluorescence and an increase in total β-III tubulin fluorescence after adding the peptide PGP; the β-III tubulin/GFAP ratio significantly increased on the 3rd day after the scratch, i.e., in the acute period of mechanical injury. The decrease in β-III tubulin levels on the 7th day was associated with the process of neuroregeneration and active migration of cells to the scratch area. Data were acquired using an inverted fluorescence microscope Zeiss Axiovert-200 (Carl Zeiss, Jena, Germany), with objective 40×/NA = 1.35 (oil) and MetaFluor software the version 7.7.0.0 (Molecular Devices, San Jose, CA, USA acquiring images of the entire observed area. The recorded images were processed in MetaFluor Analyst software the version 1.0.93 (Molecular Devices, San Jose, CA, USA) and have been converted into a Microsoft Excel table (version 2016) with total values of fluorescence intensity in each frame. In one confocal dish, six to eight photographs were taken (on both sides of the scratch in random places). In each group, there were three confocal dishes. Thus, at least 18 photographs were taken in each group. Data were analyzed using the ordinary two-way ANOVA with Sidak’s multiple comparisons test. The normality of distribution was assessed using the D’Agostino & Pearson test. Data are presented as the mean ± SD. The results were considered statistically significant with *p* < 0.05. Representative images were acquired using an LSM 880 scanning laser confocal microscope equipped with an AiryScan module and GaAsP detector (Carl Zeiss, Jena, Germany) with a Plan-Apochromat 40×/1.2 mm Corr DIC M27 multi-immersion objective. The ZEN Black tile scanning function was used to stitch four separate images in each panel. A scale bar is shown in the images. The yellow double arrow corresponds to the width of the scratch.

**Figure 4 ijms-25-10886-f004:**
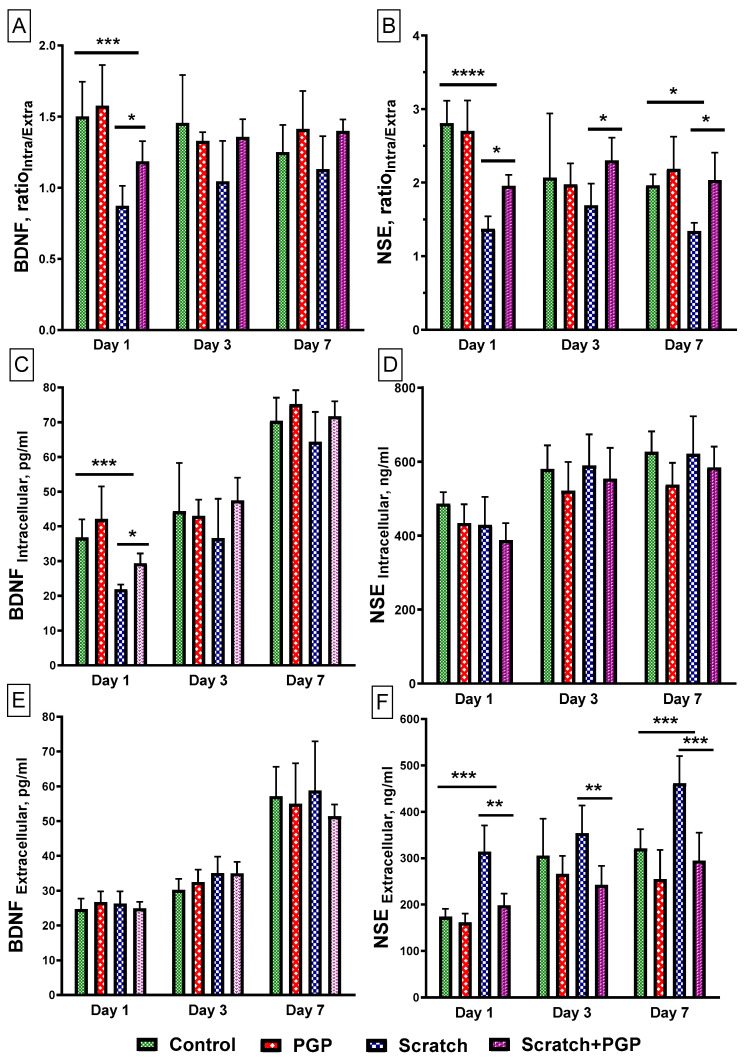
Determination of neuromarkers BDNF and NSE in a primary culture of cortical neurons after mechanical injury. The cell cultures were injured on the 4th day in culture. The number of cells in each well was 5 × 10^5^. (**A**,**B**) The ratio of intracellular to extracellular protein levels. (**C**,**D**) The intracellular protein content. (**E**,**F**) The extracellular protein content. Data are presented as mean with SD, pg/mL (BDNF) or ng/mL (NSE). Four 48-well plates with non-sister cultures were used. Each point is 5–6 wells in the plate. Total 18–22 points in each group. Statistically significant differences were detected using two-way ANOVA with subsequent Tukey’s multiple comparisons test. * *p* < 0.05; ** *p* < 0.01; *** *p* < 0.005; **** *p* < 0.0001.

**Figure 5 ijms-25-10886-f005:**
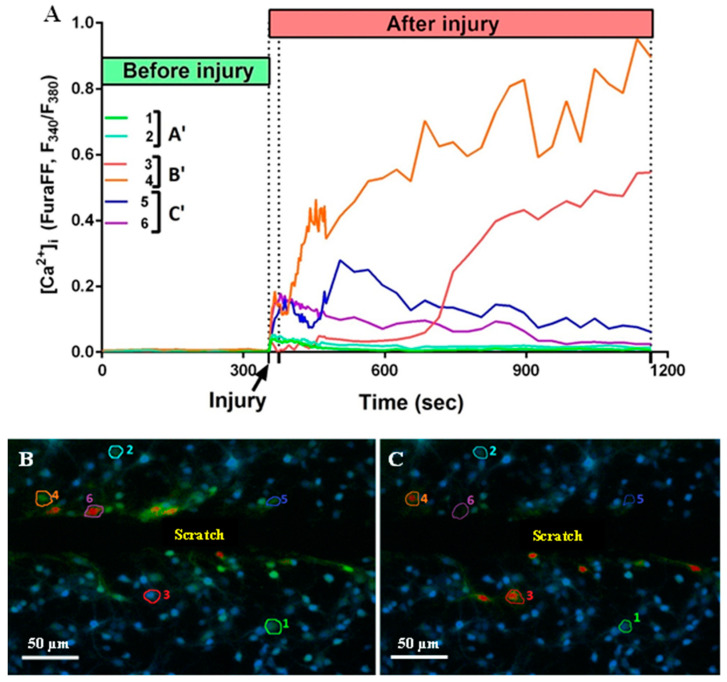
(**A**) Dynamics of changes in fluorescence of Fura-FF cells (1–6) located at different distances from the scratch. A’—uninjured cells with no Ca^2+^ dysregulation; B’—uninjured cells with developed Ca^2+^ dysregulation; C’—injured cells. (**B**) A picture of the culture 21 s after the injury. (**C**) A picture of the culture 810 s after the injury. The numbers indicate the sectors in which the cells are located.

**Figure 6 ijms-25-10886-f006:**
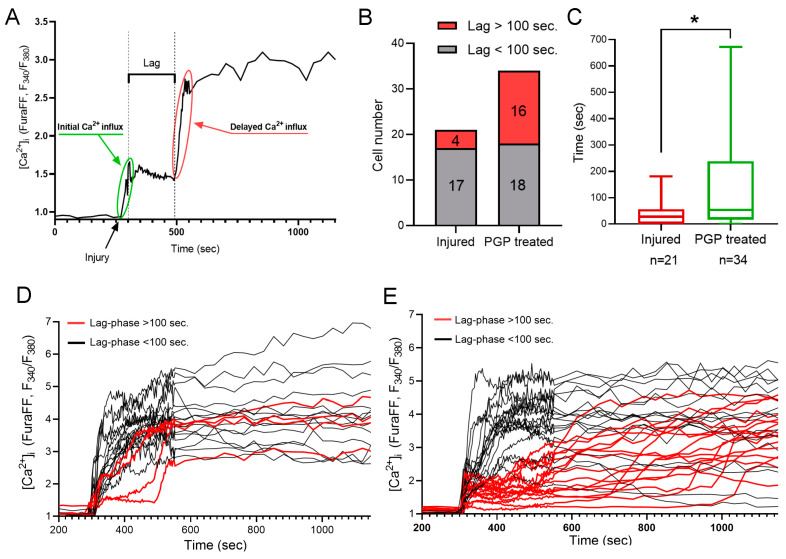
Calculation of a lag phase for a delayed Ca^2+^ influx (**A**). (**B**) Number of cells with lag over and under 100 s in untreated cultures and cultures treated with PGP. (**C**) Increased lag phase of a delayed Ca^2+^ influx after PGP addition. Dynamics of the acute changes in calcium homeostasis upon mechanical injury among cells from untreated cultures (**D**) and cultures treated with PGP (**B**). (**C**) Calculation of a lag phase for a delayed Ca^2+^ influx. (**E**). Data are presented as the median ± interquartile range. Nonparametric Mann–Whitney Test. n—the number of cells, * *p* < 0.05.

**Figure 7 ijms-25-10886-f007:**
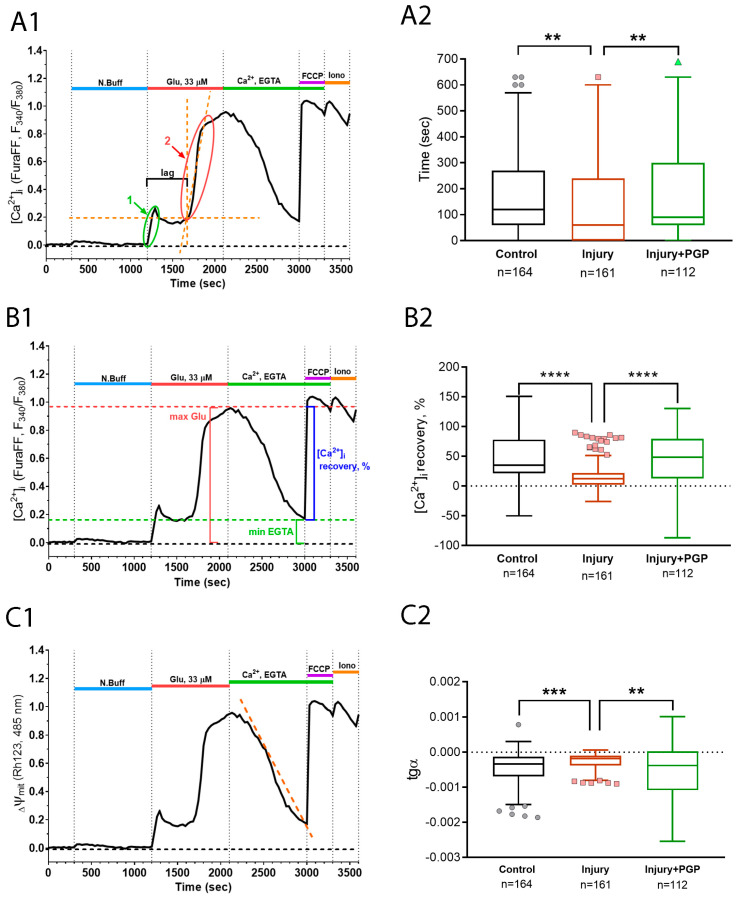
Calculation of the parameters (**A1**,**C1**) and dynamics of the delayed changes of calcium dysregulation (**A2**,**C2**) during the experiment with scratch and Glu (100 μM). (**A1**,**A2**) The lag periods of DCD in cultures; (**B1**,**B2**) recovery of [Ca^2+^]_i_ to the background level during the administration of EGTA-solution; (**C1**,**C2**) the slope of the linear approximation of the Rh123 fluorescence curve during the administration of glutamate. Non-parametric one-way analysis of variance (Kruskal–Wallis test) adjusted for multiple comparisons (Dunn’s multiple comparisons test). ****—*p* < 0.0001; ***—*p* < 0.005; **—*p* < 0.01; n—the number of cells in the group. Data (**A2**,**C2**) are presented as Tukey box plots after removing outlier values using the ROUT method with Q = 0.1% cutoff.

**Figure 8 ijms-25-10886-f008:**
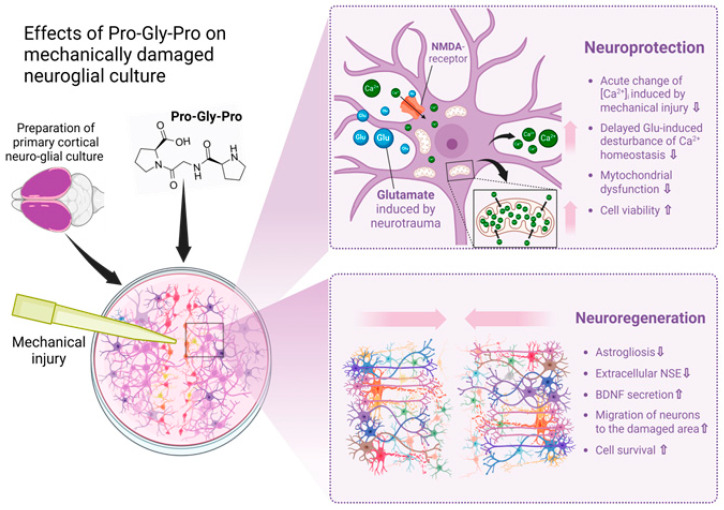
Effects of Pro-Gly-Pro peptide on neuroregeneration in a scratch test. Created in BioRender.com. The cell scratch assay showed the neuroregenerative properties of PGP, as well as its ability to protect neuro-glial cells from secondary processes that occur immediately following mechanical damage of the cell cultures. Its neuroprotective and neuroregenerative potential may be due to several effects of the peptide, which are collectively illustrated in this Figure. In the context of traumatic brain injury therapy, such an additional treatment may prevent the sequential loss of neurons and the development of post-traumatic syndrome.

## Data Availability

Data are contained within the article.

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
