# Peer review of "Regulatory Peptide Pro-Gly-Pro Accelerates Neuroregeneration of Primary Neuroglial Culture after Mechanical Injury in Scratch Tests"

_ijms, 2024, doi:10.3390/ijms252010886_

Round 1
Reviewer 1 Report (Previous Reviewer 2)
Comments and Suggestions for Authors
Review on the manuscript of Bakaeva Z et al., (ijms-3233620): “Regulatory peptide Pro-Gly-Pro accelerates neuroregeneration of primary neuroglial culture after mechanical injury in scratch-test”.
In this study, the Authors explored the protective potential of the Pro-Gly-Pro (PGP) peptide in an in vitro model of traumatic brain injury (TBI) induced by scratching. The Authors show that applying a scratch to cortical cultures resulted in neuronal death, decreased expression of BDNF, increased release of NSE, and disruption of calcium homeostasis and mitochondrial membrane potential. These effects were attenuated in cells treated with PGP. Additionally, scratch application accentuated the disruption of calcium homeostasis and mitochondrial membrane potential caused by a subtoxic dose of glutamate, effects also attenuated by PGP. Therefore, the peptide PGP exhibited a neuroprotective effect by enhancing the survival of neuroglial cells following mechanical trauma in vitro. It achieves this by reducing cellular calcium overload and preventing mitochondrial dysfunction.
Overall, I find this topic to be of great interest, as studying TBI is crucial due to its potential to cause severe and long-lasting physical, cognitive, and emotional impairments, which affect a person's quality of life and ability to function. Substances that positively impact intracellular calcium buffering systems or cellular energy supply could potentially reduce cell death resulting from TBI. I believe the Authors have addressed the main question proposed. The manuscript is well-written and well-organized, demonstrating rigor in presenting the results. The issues I have identified with the current form of the manuscript are listed below. I hope the Authors find the following comments and suggestions useful.
1 - I recommend that the Authors correct the information in the abstract: “profile of neuronal markers (BDNF, NSE and GFAP)”. From what I know, GFAP is a marker of astrocytes.
2 - In figure 2, I recommend that the authors include a panel with the cells stained with Hoechst 33342. It is important to mark the entire cell population in that area, as some cells may be dead and, therefore, not visualized with Syto-13 staining.
3 - As a proof-of-concept for the experiment shown in Figure 7, it would be interesting to test NMDA receptor inhibition with MK-801. In principle, this should prevent the effects of glutamate. Additionally, it would be interesting to see whether the PGP peptide and MK-801 prevent cell death caused by glutamate. This way, the hypothesis would be perfectly clarified.
4 - In subsection 4.9, Statistical Analysis, some corrections are needed. The Authors state:
- “Statistical data analysis was performed with the aid of GraphPad Prism 6 (GraphPad Software Inc., USA),” but later state that “Statistical analysis was performed using GraphPad Prism 8 (GraphPad Software, San Diego, CA, USA).”
- “All experiments were performed in duplicate at least 4 times on non-sister cell cultures,” but later mention that “At least three independent experiments were conducted in each series.”
- “Normality of distribution was assessed using the Shapiro-Wilk test,” but later state that “The data were checked for normality using three tests: D'Agostino-Pearson omnibus normality test, Shapiro-Wilk normality test, and the KS normality test.”
Therefore, these inconsistences need clarification.
- 5 - Technically, it is difficult to understand how applying mechanical injury during calcium imaging does not move the slide in the microscope. Could the authors clarify this methodological question?
Author Response
We thank Reviewer for their interest in the article and its careful analysis. All received comments and remarks allowed us to improve the article. All corrections were followed in Word in the review mode in color. We also thank Reviewer for the high evaluation of the work.
Comment 1: I recommend that the Authors correct the information in the abstract: “profile of neuronal markers (BDNF, NSE and GFAP)”. From what I know, GFAP is a marker of astrocytes.
Response 1: Thanks for the note! The information in the annotation has been corrected
Comment 2: In figure 2, I recommend that the authors include a panel with the cells stained with Hoechst 33342. It is important to mark the entire cell population in that area, as some cells may be dead and, therefore, not visualized with Syto-13 staining.
Response 2: As can be seen in Figure 1D, and even better in our other article DOI: 10.3390/ijms23073858 (Figure 1D), in the injured area immediately after the scratch there are almost none dead cells and not at all living cells, due to their complete erasure. Accordingly, the cells stained by (Syto-13) and displayed in Figure 2 (D-E) are the living cells who have migrated into the injured area in the 7th day after the scratch. The quantitative representation of this process is shown In Figure 2 (C).
Comment 3: As a proof-of-concept for the experiment shown in Figure 7, it would be interesting to test NMDA receptor inhibition with MK-801. In principle, this should prevent the effects of glutamate. Additionally, it would be interesting to see whether the PGP peptide and MK-801 prevent cell death caused by glutamate. This way, the hypothesis would be perfectly clarified.
Response 3: You are absolutely right! Previously, we demonstrated the involvement of NMDA receptors in processes leading to delayed neuronal death due to calcium dysregulation and synchronous mitochondrial depolarization (DOI: 10.3390/ijms23073858). Mechanical injury to the primary neuroglial culture in the form of a scratch caused acute disruption of calcium homeostasis and mitochondrial functions. This was accompanied by neuronal death. An increase in [Ca2+]i and a decline in ΔΨm were observed ~10 s after the injury in cells that were located not further than 150-200 µm from the scratch boundary. The entry of Ca2+ into cells during mechanical damage to the primary neuroglial culture occurred predominantly through the NMDA-type of the glutamate ionotropic channels. MK801, an inhibitor of this type of the glutamate receptors, prevented the acute increase of the [Ca2+]i in 99% of the neurons. However, pathological changes in calcium homeostasis persisted in the primary neuroglial culture for a week after the injury. In this work, we studied the peptide's potential to eliminate or minimize these pathological processes. As for the direct effects of PGP on Glu receptors, we are currently studying this issue carefully and plan to describe the results in our next article.
Comment 4: In subsection 4.9, Statistical Analysis, some corrections are needed. The Authors state:
- “Statistical data analysis was performed with the aid of GraphPad Prism 6 (GraphPad Software Inc., USA),” but later state that “Statistical analysis was performed using GraphPad Prism 8 (GraphPad Software, San Diego, CA, USA).”
- “All experiments were performed in duplicate at least 4 times on non-sister cell cultures,” but later mention that “At least three independent experiments were conducted in each series.”
- “Normality of distribution was assessed using the Shapiro-Wilk test,” but later state that “The data were checked for normality using three tests: D'Agostino-Pearson omnibus normality test, Shapiro-Wilk normality test, and the KS normality test.”
Response 4: Thanks for the note! All inconsistencies in the description of statistical analysis were clarified. However, there were some discrepancies in the description of the selected test for the analysis of the reliability of differences, as it depended on the design of the experiment. Also, the choice of the normality test was related to the number of n and the test proposed by the program GraphPad Prism 8 itself.
Comment 5: Technically, it is difficult to understand how applying mechanical injury during calcium imaging does not move the slide in the microscope. Could the authors clarify this methodological question?
Response 5: Indeed, there is a risk of losing focus, and it is also challenging to adjust the needle pressure on the cell culture to avoid detaching the entire culture from the substrate. However, if the confocal dish is well-secured and the pressure is properly adjusted, it is possible to create a scratch directly under the microscope objective during an experiment to measure calcium levels. This was done as follows.
Petri dish was fixed in place with 2 metal slide holders. Threaded pins were used to adjust the tension with which the dish was fixed. Later we focused the microscope on a neuroglial culture. Next step was to lower the needle connected to the micromanipulator the way that it would only touch the culture out of the acquisition focus. After making sure that the needle was touching the culture, but not disrupting the microscope focus, the scratch was applied the way it would go through the center of focus. This procedure is laborious, but after multiple attempts we were able to find a balance between the force of the needle pushing on the culture and holders, fixing the petri dish in place, so scratching the culture wouldn’t disrupt the focus or move the dish. This was a difficult challenge we faced, but we managed it. Thanks for the comment!
P.S. A mechanical micromanipulator with a 2 ml syringe needle set at an angle of 45 degrees was used. Our micromanipulator is equipped with two microscrews, one of which is used to adjust the descent of the needle onto the dish. Before the experiment, the needle was touched to the glass in a cell-free zone under the microscope, consistently setting the same mark on the micromanipulator handle in all experiments. In this way, we adjusted the vertical position of the needle. The other microscrew was used to form a scratch, i.e., to move the needle horizontally during the experiment. The scratch was in the field of view of the fluorescence microscope camera. This ensured that the pressure on the glass was always the same.
Reviewer 2 Report (Previous Reviewer 1)
Comments and Suggestions for Authors
The authors did a good job of addressing the concerns of the reviewers.
There are still a couple of minor points:
1. In Figure 2B, the authors reported the ration of live / dead cells, but did not show images of the EthD staining. They state in their rebuttal that this is because there was non staining. If so, how are you obtaining a ratio?
2. Figure 3F- why does the tubulin staining decrease so much between Day 3 and 7 in the PGP treated cells? This seems to contradict a compensatory migration into the wound area- at least one that is lasting for regeneration.
3. Figure 7A2- I still take issue with the statistics on some of the figures, despite the reported values being medians +/- max and min. This is particularly true on Fig. 7A2, which shows almost complete overlap of all three conditions.
Author Response
We are very grateful to Reviewer for the interest in our study and for valuable comments that made it possible to improve the text of the manuscript. Please note that we have updated Figure 6 and Figure 7 and added changes to the legend. All corrections were followed in Word in the review mode in color. Please note that we have updated Figure 6 and Figure 7 and added changes to the legend.
Comment 1: In Figure 2B, the authors reported the ration of live / dead cells, but did not show images of the EthD staining. They state in their rebuttal that this is because there was non staining. If so, how are you obtaining a ratio?
Response 1: Please note, in Figure 2B shows Live/Dead cells ratio (Syto-13/EthD-1) in areas which are located near the scratch and not in the scratch itself. The microphotographs of the areas near the scratch are shown in Figure 1. And there is EthD-1 staining there.
The microphotographs of the scratch areas You can see in the Figure 2D and Figure 2E (after treatment with PGP). Figure 2C shows the number of living cells (Syto-13) invaded in scratch area. There are no dead cells, since all the cells were erased by the scratch, and those that appeared on the 3rd and 7th days were those that migrated to the damaged area.
Comment 2: Figure 3F- why does the tubulin staining decrease so much between Day 3 and 7 in the PGP treated cells? This seems to contradict a compensatory migration into the wound area- at least one that is lasting for regeneration.
Response 2: This is a very interesting question! The thing is that the ratio of β-III tubulin fluorescence to Hoechst 33342 fluorescence (Figure 3F) was constructed from fluorescent photographs taken on both sides of the scratch (described in lines 168-170). Therefore, the decrease in b3-tubulin in the scratched cell culture between days 3 and 7 may indicate a gradual migration of neurons directly to the scratch area, as evidenced by the results in Figures 2C and 2E. In simple terms, the number of neurons in the scratch area increases, while in the areas on both sides of the scratch it decreases. However, you may have noticed that b3-tubulin also decreases in the untreated group, which also indicates a migration process, although it is much less than with the addition of PGP. Notice there are so many nuclei on the border with the scratch after adding PGP.
Comment 3: Figure 7A2- I still take issue with the statistics on some of the figures, despite the reported values being medians +/- max and min. This is particularly true on Fig. 7A2, which shows almost complete overlap of all three conditions.
Response 3: After removing outlier values from all the groups (ROUT method, Q=0.1% cutoff), the statistical significance got even more pronounced, as you can appreciate on the updated graphs (Figures 7 A2-C2). Data was non-parametrically distributed (according both the Shapiro-Wilk and Kolmogorov-Smirnov tests), therefore – Kruskal-Wallis with Dunn’s multiple comparison correction was applied, which is true for Fig7 A2-C2. This time we used Tukey box plots to represent our data. For the sake of this discussion, here is the same Fig7 A2 graph represented as individual values. A large number of n. (Please see attachment)
As you can see, both control and “Injury + PGP” groups have these clusters of cells with especially long lag-phase of secondary Ca2+ influx, which is not that pronounced in “Injury” group. These “resistant” neurons were equally distributed between different experiments we performed and in no way pose an experimental artefact.
We think that presence of PGP allows to maintain this population of “resistant” neurons that would otherwise be lost in course of mechanical injury. This hypothesis is for now only a matter of speculation, and we don’t know what would be a physiological mechanism underlying an observed effect, but we hope this helped to clarify the situation a little.

This manuscript is a resubmission of an earlier submission. The following is a list of the peer review reports and author responses from that submission.
Round 1
Reviewer 1 Report
Comments and Suggestions for Authors
This paper uses a cell culture model that aims to use mechanical stress in order to model TBI, then to determine the efficacy of a peptide treatment to ameliorate detrimental effects on the neural culture. While the general premise is straightforward, there are several issues that dampen enthusiasm for the manuscript:
1. As always, with cell culture experiments, you lose involvement of systemic factors in the effects of treatment. Is this peptide bioavailable? Does it cross the blood- brain barrier? Does it affect other organ systems in a way that could affect the brain downstream? Are the concentrations in used cell culture representative of what would be found in vivo in the brain?
2. All of the cited literature on PGP is from the same lab or labs. Is there no other work on this peptide? In the same vein, it looks like the same authors have published similar work in a primary neuroglial culture of glutamate toxicity using PGP. I do not have access to this journal, but it looks like the experiments are similar, but with the absence of the scratch component. That makes this work seem a bit incremental.
3. There are a couple of places where there should be additional representative panels of the IF staining- Figure 2 does not show the EthD-1 staining that was used in the quantitation.
4. In Figure 3, there is no quantitation of GFAP or tubulin staining, just a single representative image of each treatment. There could be well to well variability in the number of astrocytes and neurons in and around the scratch that even out upon quantitation of multiple wells. There is no way to tell from a single image. There is very little Hoescht staining in Figure 3B, which may be accounted for by variation, but this cannot be evaluated.
5. The number of replicates, and what is counting as replicates, is not always clear. The authors state that each experiment was repeated at least 3 independent times, but how is that accounted for in the statistics? Are there replicates within each independent experiment? Are those replicates accounted for differently than the independent repeats? Figure 2 does not list an N. The data for Figure 6-7 list N as the number of cells. Are all of the cells in the same well, same set of replicates, or across all replicates?
6. Based on the errors noted, it does not look like some of the data is significantly different, though it is denoted as such. This is especially true for Figures 6-7, the data for which has very large standard deviations.
7. There is no quantitation for Figure 5 and it is difficult to see some of the cells the authors are trying to draw attention to.
8. The authors need to be careful not to overstate the conclusions based on the data. They refer to changes in the neuronal network, but only show changes in neuronal staining, not network connectivity. This is just one example.
Comments on the Quality of English LanguageOverall, the English was understandable. There were a few minor grammatical errors that could be improved.
Reviewer 2 Report
Comments and Suggestions for Authors
Review on the manuscript of Bakaeva Z et al.: “Regulatory peptide Pro-Gly-Pro accelerates neuroregeneration of primary neuroglial culture after mechanical injury in scratch-test”.
In this study, the Authors explored the protective potential of the Pro-Gly-Pro (PGP) peptide in an in vitro model of traumatic brain injury (TBI) induced by scratching. The Authors show that applying a scratch to cortical cultures resulted in neuronal death, decreased expression of BDNF, increased release of NSE, and disruption of calcium homeostasis and mitochondrial membrane potential. These effects were attenuated in cells treated with PGP. Additionally, scratch application accentuated the disruption of calcium homeostasis and mitochondrial membrane potential caused by a subtoxic dose of glutamate, effects also attenuated by PGP. Therefore, the peptide PGP exhibited a neuroprotective effect by enhancing the survival of neuroglial cells following mechanical trauma in vitro. It achieves this by reducing cellular calcium overload and preventing mitochondrial dysfunction."
Overall, I find this topic to be of great interest, as studying TBI is crucial due to its potential to cause severe and long-lasting physical, cognitive, and emotional impairments, which affect a person's quality of life and ability to function. Substances that positively impact intracellular calcium buffering systems or cellular energy supply could potentially reduce cell death resulting from TBI. I believe the Authors have addressed the main question proposed. The manuscript is well-written and well-organized, demonstrating rigor in presenting the results. The issues I have identified with the current form of the manuscript are listed below. I hope the Authors find the following comments and suggestions useful.
1 - I recommend that the Authors correct the information in the abstract: “profile of neuronal markers (BDNF, NSE and GFAP)”. From what I know, GFAP is a marker of astrocytes.
2 - In figure 2, I recommend that the authors include a panel with the cells stained with Hoechst 33342. It is important to mark the entire cell population in that area, as some cells may be dead and, therefore, not visualized with Syto-13 staining.
3 - I recommend that the authors improve the quality of the figures, particularly Figure 3.
4 - For the graphs in Figure 4, I recommend that the authors use different colors for the different conditions. The gray scale makes them difficult to interpret.
5 - For BDNF, the authors quantified its intracellular levels. However, for NSE, they quantified the extracellular levels. I recommend that the authors show both the intracellular and extracellular levels for both markers to make the conclusion clearer. Additionally, is there any specific reason for BDNF to be reduced in the intracellular compartment, whereas NSE increases in the extracellular compartment?
6 - As a proof-of-concept for the experiment shown in Figure 7, it would be interesting to test NMDA receptor inhibition with MK-801. In principle, this should prevent the effects of glutamate. Additionally, it would be interesting to see whether the PGP peptide and MK-801 prevent cell death caused by glutamate. This way, the hypothesis would be perfectly clarified.
Minor points:
1 - On line 369, could the authors clarify the duration of the centrifugation?
2 - On line 436, the information is repeated. Can the Authors correct it?
3 - Could the Authors clarify what M ± m means?
4 – Could the Authors clarify what they mean with “Based on the obtained results, a criterion was selected for further statistical processing”.
5 - Technically, it is difficult to understand how applying mechanical injury during calcium imaging does not move the slide in the microscope. Could the authors clarify this methodological question?